# On the Limitations of Visual-Semantic Embedding Networks for Image-to-Text Information Retrieval

**DOI:** 10.3390/jimaging7080125

**Published:** 2021-07-26

**Authors:** Yan Gong, Georgina Cosma, Hui Fang

**Affiliations:** Department of Computer Science, School of Science, Loughborough University, Loughborough LE11 3TT, UK; h.fang@lboro.ac.uk

**Keywords:** visual-semantic embedding network, multi-modal deep learning, cross-modal, information retrieval

## Abstract

Visual-semantic embedding (VSE) networks create joint image–text representations to map images and texts in a shared embedding space to enable various information retrieval-related tasks, such as image–text retrieval, image captioning, and visual question answering. The most recent state-of-the-art VSE-based networks are: VSE++, SCAN, VSRN, and UNITER. This study evaluates the performance of those VSE networks for the task of image-to-text retrieval and identifies and analyses their strengths and limitations to guide future research on the topic. The experimental results on Flickr30K revealed that the pre-trained network, UNITER, achieved 61.5% on average Recall@5 for the task of retrieving all relevant descriptions. The traditional networks, VSRN, SCAN, and VSE++, achieved 50.3%, 47.1%, and 29.4% on average Recall@5, respectively, for the same task. An additional analysis was performed on image–text pairs from the top 25 worst-performing classes using a subset of the Flickr30K-based dataset to identify the limitations of the performance of the best-performing models, VSRN and UNITER. These limitations are discussed from the perspective of image scenes, image objects, image semantics, and basic functions of neural networks. This paper discusses the strengths and limitations of VSE networks to guide further research into the topic of using VSE networks for cross-modal information retrieval tasks.

## 1. Introduction

Visual-semantic embedding (VSE) networks jointly learn representations of images and texts to enable various cross-modal information retrieval-related tasks, such as image–text retrieval [1,2,3,4], image captioning [5,6,7,8], and visual question answering (VQA) [4,9,10]. Although great progress has been made in the field of VSE networks for information retrieval, cross-modal information retrieval remains a challenging problem because of the abundant information contained in visual semantics [3].

The state-of-the-art VSE networks are: VSE++ [1], SCAN [2], VSRN [3], and UNITER [4]. To tackle cross-modal information retrieval using VSE-based networks, Faghri et al. [1] proposed a hard negative loss function as part of VSE++, for the network to learn to make the relevant target closer to the query than other items in the corpus. Lee et al. [2] applied the stacked cross attention (SCAN) mechanism to align image regions and words for improving VSE networks. Li et al. [3] proposed the visual semantic reasoning network (VSRN) for extracting high-level visual semantics with the aid of the graph convolution network (GCN) [11]. Chen et al. [4] introduced a pre-trained network with the transformer [12], namely the universal image-text representation (UNITER), to unify various cross-modal tasks such as VQA and image–text matching.

Existing literature on VSE networks provides an evaluation of the performance of these networks on benchmark datasets such as MSCOCO [13] and Flickr30K [14], but to the best of the authors’ knowledge, there are no papers that provide an in-depth investigation of the performance of the methods and their limitations. Some of the known limitations are that the network architectures of VSE++ and SCAN have not been specifically designed to extract high-level visual semantics [3]. Additionally, VSRN lacks a suitable attention mechanism [4] to align the image and text in the same latent space, and UNITER requires large amounts of data for pre-training [15]. However, it is important to understand the limitations of these networks in order to devise strategies for mitigating these limitations when developing new and improved VSE networks and/or when extending the capabilities of existing ones. Therefore, this study aims to identify, investigate, and classify the limitations of state-of-the-art VSE networks (VSE++, SCAN, VSRN, and UNITER) when they are applied to information retrieval tasks.

This paper is organised as follows. Section 2 provides an overview of VSE networks. Section 3 discusses the dataset, evaluation measures, and experiment methodologies. Section 4 presents the results of VSE++, SCAN, VSRN, and UNITER, summarises the limitations of VSE networks, and discusses their strengths and limitations to guide further research. Section 5 provides a conclusion and future work.

## 2. Related Methods

VSE networks aim to align representations of relevant images and descriptions in the same latent space for cross-modal information retrieval. As shown in Figure 1, a typical VSE network embeds features of image regions of interest (ROIs) and word vectors of descriptions. Where ROIs are analysed by object detection models [16,17,18], faster R-CNN [16] is commonly used by VSE networks, including VSE++, SCAN, VSRN, and UNITER.

For image-to-text retrieval, the VSE network finds the most relevant descriptions to the image query by ranking the similarity scores between the image query’s embedding and a set of description embeddings. For text-to-image retrieval, the VSE network finds the most relevant images to the description query by ranking the similarity scores between the description query’s embedding with a set of image embeddings.

In the literature, VSE++ [1] was a milestone for VSE networks. It deployed a fully connected neural network to extract image features of ROIs obtained by a faster R-CNN [16] and a gated recurrent unit (GRU) network [19] to embed textual features. In addition, online hard-negative mining was proposed to improve the network’s convergence of this joint-embedding space [1]. In particular, the hardest negative triplet loss (MH) function defined in Formula (Equation 1) was used in the network training phase.
(1)$$\begin{array}{c} l_{MH}\left(i,d\right)=\max_{\hat{d}}{\left[\alpha +s\left(i,\hat{d}\right)-s\left(i,d\right)\right]}_{+}+\max_{\hat{i}}{\left[\alpha +s\left(d,\hat{i}\right)-s\left(i,d\right)\right]}_{+} \end{array}$$
where α is a constant parameter of margin, and ${\left[x\right]}_{+}\equiv \max\left(x,0\right)$. Let *i* and *d* denote an image and its relevant (i.e., corresponding) description, respectively. Let $d^{{}^{\prime }}$ denote a description irrelevant to image *i*, and $i^{{}^{\prime }}$ denote an image irrelevant to description *d*. Let s(i,d) be the similarity score between a relevant image *i* and description *d*; $s(i,\hat{d})$ be the similarity score of an image *i* and an irrelevant description $d^{{}^{\prime }}$; and $s(d,\hat{i})$ be the similarity score of a description *d* and an irrelevant image $i^{{}^{\prime }}$. Given a relevant image–description pair (i,d), the result of the function only takes the maximum values of irrelevant pairs $s(i,d^{{}^{\prime }})$ and $s(d,\hat{i})$. MH loss avoids local minima because it mainly focuses on the hardest negative, the winner takes all gradients [1].

Attention-based models have been exploited to improve the performance of VSE++. SCAN [2] was proposed to emphasise salient regions and keywords for the alignment when building the embedding space. The importance of each image region is learned based on the similarity between individual attended sentence word vectors and image region features. Wang et al. [15] proposed the position attention network (PFAN++) that utilises the position-focused attention module to enhance image feature representations of visual semantics. Recently, the GCN has also been utilised in VSE networks by VSRN [3]. VSRN [3] was proposed to explore object spatial relationships in an image to strengthen image feature representations. After identifying ROIs in the image, the relationship between each region is learned using (Equation 2) for feature enhancement.
(2)$$\begin{array}{c} R=\varphi {\left(V\right)}^{T}\phi \left(V\right) \end{array}$$
where *R* represents the correlation between a set of image features denoted as a matrix *V*, the weight parameters φ and ϕ can be learnt through back propagation. Then, the residual connection to the GCN is added for enhancement based on Formula (Equation 3).
(3)$$\begin{array}{c} V^{*}=W_{r}\left(RVW_{g}\right)+V \end{array}$$
where $W_{g}$ is the weight matrix of the GCN layer, $W_{r}$ is the weight matrix of the residual structure, and *R* is the affinity matrix from Formula (Equation 2), so the output $V^{*}$ is the relation-enhanced representation of image regions. In addition, VSRN further connected the output of GCN with a GRU [19] for global visual semantic reasoning.

The transformer architecture is emerging in VSE networks since it has achieved great success in natural language processing [12]. UNITER [4] adopts a transformer model for joint multi-modal embedding learning. The model of UNITER was pre-trained with four tasks, including masked language modeling (MLM), masked region modeling (MRM), image–text matching (ITM), and word–region alignment (WRA), on large amounts of data. These tasks facilitate the creation of a generalisable embedding space with contextual attention on both image and textual features as well as modeling their joint distribution.

## 3. Materials and Methods

Image-to-text retrieval refers to the task of retrieving relevant image descriptions when given an image as a query. This section describes two experiments. Experiment 1 evaluates the performance of state-of-the-art VSE networks (VSE++ [1], SCAN [2], VSRN [3], and UNITER [4]) for the task of image-to-text retrieval using the Flickr30k dataset. Experiment 2 provides a limitation analysis of the performance of two networks, VSRN [3] and UNITER [4]. Note that for experiment 2, VSRN and UNITER were selected because they outperformed other networks based on the results of experiment 1 (Section 4.1) and because recent comparisons found in the literature [3,4] also show those to be the best-performing VSE networks.

### 3.1. Dataset and VSE Network Preparation

Flickr30K [14], a benchmark dataset that is typically used for evaluating the performance of deep learning models, was utilised for the experiments. Every image in the Flickr30K dataset is associated with five relevant textual descriptions, as shown in Figure 2. Flickr30K was split into training, validation, and test sets containing 29,000, 1014, and 1000 images, respectively [7].

For a fair comparison of VSE++, SCAN, VSRN, and UNITER, the following adaptations were made. For VSE++ [1], the backbone of the faster R-CNN object detection method was changed from VGG19 [20] to ResNet-101 [21] to be consistent with SCAN, VSRN, and UNITER. For SCAN [2], image-to-text LogSumExp (i-t LSE) pooling was adopted because it was found to be the most suitable model setting for image-to-text attention models when tested on Flickr30K [2]. UNITER’s image–text matching (ITM) model was selected for the task. No adaptations were made to VSRN [3], or to UNITER’s ITM model. Note that UNITER’s [4] ITM model has been pre-trained on four large datasets, namely, MS COCO [13], Visual Genome [22], Conceptual description [23], and SBU captions [24]. VSE++, SCAN, and VSRN have not been pre-trained on other datasets.

### 3.2. Performance Evaluation Measures

The measures adopted for evaluating the image–text retrieval performance of the VSE networks are Recall, Precision, $F_{1}$-score, and interpolated Precision–Recall PR curves. These are described below.

Recall is the percentage of relevant textual descriptions retrieved over the total number of textual descriptions relevant to the query. Recall is computed using (Equation 4).
(4)$$Recall=\frac{Total\;number\;of\;relevant\;textual\;descriptions\;retrieved}{Total\;number\;of\;relevant\;textual\;descriptions}$$

Precision is a percentage of relevant textual descriptions retrieved over the total number of textual descriptions retrieved. Precision is computed using (Equation 5).
(5)$$Precision=\frac{Total\;number\;of\;relevant\;textual\;descriptions\;retrieved}{Total\;number\;of\;textual\;descriptions\;retrieved}$$

$F_{\beta }$-score combines the results of Recall and Precision using (Equation 6):(6)$$\begin{array}{c} F_{\beta }=\frac{(\beta^{2}+1)P*R}{\beta^{2}\left(P+R\right)} \end{array}$$

The $F_{\beta }$-score can be adjusted to allow for weighting Precision or Recall more highly. As the importance of Recall and Precision is equal in the image–text retrieval task, the experiment sets the parameter β=1 as shown in (Equation 7):(7)$$\begin{array}{c} F_{1}=\frac{2*P*R}{P+R} \end{array}$$

A PR curve is a plot of the Precision (y-axis) and the Recall (x-axis) for different thresholds. An interpolated PR curve [25] shows Precision (*P*) interpolated for each standard Recall (*R*) level as shown in (Equation 8).
(8)$$R\in \left\{0.0,0.1,0.2,0.3,0.4,0.5,0.6,0.7,0.8,0.9,1.0\right\}.$$

Specifically, $P(R_{j})$, the maximum Precision at any Recall between the *j*th and (j+1)th level is taken to interpolate for $R_{j}$, shown in (Equation 9).
(9)$$P\left(R_{j}\right)=\max_{R_{j}<R<R_{j+1}}P\left(R\right).$$

### 3.3. Experiment 1 Methodology for the Comparison of VSE Networks for Image-to-Text Retrieval

Experiment 1 compares the image-to-text retrieval performance of VSE++, SCAN, VSRN, and UNITER for 1000 image queries from Flickr30K’s test set. As previously mentioned, every Flickr30K image has five textual descriptions, and therefore there are 5000 descriptions in the query set (1000 images × 5 descriptions per image = 5000 descriptions).

Given a query, the top *n* image descriptions that are relevant to a query are retrieved. Then, the retrieval performance of each model is evaluated based on three strategies: the relevance of the first retrieved description (i.e., Recall@1); whether any 1 of the 5 descriptions are retrieved in the top *n* results (i.e., Recall@5, @10, @20), which is the same evaluation strategy followed by [1,2,3,4]; and the retrieval performance of the model with regard to retrieving all 5 relevant descriptions when looking at the top *n* retrieved descriptions (i.e., Recall, Precision, $F_{1}$-score @5, @10, @20, @50, @100), which is a tougher strategy than the one followed by [1,2,3,4].

Furthermore, the performance of the algorithms in retrieving all 5 descriptions is evaluated for two main reasons: (1) because the ability to retrieve all relevant descriptions is important for information retrieval (IR) systems [26] and (2) given that the study focuses on the analysis of limitations of VSE networks, evaluating the performance of VSE networks using the more challenging criteria of retrieving the 5 descriptions can be more beneficial for the task of discovering the limitations of VSE networks when they are used for IR tasks.

### 3.4. Experiment 2 Methodology for Finding the Limitations of VSE Networks

Experiment 2 analyses the performance of the two VSE networks, i.e., VSRN and UNITER, that performed best in experiment 1, in order to identify their limitations. Figure 3 illustrates the methodology for experiment 2.

A description of each step is provided below:

Step 1: The query set comprises images and their corresponding relevant descriptions from the test (*n* = 1000 queries) and validation (*n* = 1014 queries) sets of Flickr30K combined into a single set containing 2014 queries.

Step 2: For the purposes of evaluating the performance of VSRN and UNITER across the different image classes, the images found in the query set were grouped into classes. Therefore, the query images were classified into 453 classes using the ImageNet [27] class labels (up to 1000) with the aid of a trained Resnet [21] model. Table 1 shows the labels of the 40 largest classes (i.e., they contain the largest number of images).

Step 3: VSRN and UNITER were evaluated on different image classes from Step 2 using the evaluation measures of average Recall@5 and average Precision@1 (see Section 3.2).

Step 4: The top 25 worst-performing classes (and their images), based on average Recall@5 results, were extracted to be used for the task of identifying the limitations of the models. Classes with fewer than 10 images were removed.

Step 5: All images with irrelevant retrieved descriptions when using the Precision@1 evaluation measure were taken and analysed manually to identify reasons that the models did not retrieve those descriptions, and to further summarise those reasons into a set of limitations.

## 4. Results

This section describes the results of experiments 1 and 2. The experimental methodology of each experiment is presented in Section 3.3 and Section 3.4, respectively.

### 4.1. Results of Experiment 1: Comparison of VSE++, SCAN, VSRN, and UNITER for Image-to-Text Retrieval

Initially, VSE++, SCAN, VSRN, and UNITER were evaluated in terms of their performance in retrieving any one of the five relevant textual descriptions for each query. Performance was averaged across all n=1000 queries to obtain the average Recall@1, @5, @10, and @20 values, as shown in Table 2. UNITER achieved the highest average Recall (i.e., average Recall@1 = 80.8%), VSRN, SCAN, and VSE++ achieved 69.3%, 67.5%, and 40.0% on average Recall@1, respectively. These results are consistent with those reported in [1,2,3,4].

Next, VSE++, SCAN, VSRN, and UNITER were evaluated in terms of their performance in retrieving all five of the relevant textual descriptions for each query. Table 3 compares the performance of the models when considering the top *K* results, where *K* is a predefined number of descriptions retrieved. The results show that UNITER consistently achieved the best performance across all evaluation measures and for all K values.

Figure 4 presents the interpolated PR curves of each image–text retrieval model. UNITER outperformed all other three networks, followed by VSRN. Figure 4 highlights that UNITER and VSRN are more effective models when considering both the Recall and Precision evaluation measures. In conclusion, the results of experiment 1 demonstrate that the best-performing image-to-text retrieval model is UNITER, followed by VSRN.

Figure 5 shows the computation time of VSE++, SCAN, and VSRN against the number of training samples from Flickr30K [14] for one epoch. The running time of each epoch was the same for each individual algorithm, and therefore for ease of comparison the computation time was calculated for one epoch for each training sample. The number of training samples increased from 2900 to 29,000 in steps of 2900. The three lines that fit the data points of VSE++, SCAN, and VSRN follow the equations of $T_{VSE++}\left(n\right)=0.0018n-0.2173$, $T_{SCAN}\left(n\right)=0.0119n+0.5239$, and $T_{VSRN}\left(n\right)=0.0159n+0.0365$, respectively. UNITER has not been included in the comparison because it was pre-trained on four other datasets [13,22,23,24], whereas the other three models, VSE++, SCAN, and VSRN, were only trained on Flickr30K. Hence, for a consistent comparison of the performance of the models, UNITER was excluded from the comparison.

### 4.2. Results of Experiment 2: Limitations of VSE Networks

This experiment concerns an analysis of the limitations of VSE networks. Only the UNITER and VSRN models were utilised for this experiment since the experiment 1 results and the literature agreed that these are the best-performing VSE models. Table 4 shows the 25 worst-performing classes for the query set using the methodology described in Section 3.4.

The results of VSRN and UNITER contain 16 identical classes, and this suggests that they share some common limitations. Focusing on these worst-performing classes, an in-depth analysis of the retrieved descriptions that are irrelevant to the image queries in these classes has revealed 10 limitations of VSE networks they and are summarised in Table 5. These limitations were generalised into four groups.

The discussion that follows refers to the limitations provided in Table 5. The limitations of group 1 occur when a VSE network does not globally understand the image scene. Limitation 1 shows that VSRN cannot specifically distinguish the importance between foreground and background information in an image. For example, in Figure 6a, people and room facilities in the background caused VSRN to misinterpret the image as ‘mopping in a subway station’ rather than ‘counting change in a store’. However, UNITER overcomes this limitation by using the self-attention mechanism of the multi-layer transformer to learn the relations between image objects. Furthermore, limitation 2 reveals an issue of VSRN and UNITER with missing key image objects in understanding the image. For example, the retrieved textual description from VSRN and UNITER for Figure 6b does not mention that a person is standing in front of the restaurant. Therefore, the background and object information in an image needs to be considered from a global angle in VSE networks.

Limitations in group 2 revealed that the VSE network does not give enough attention to the details of image objects, and hence the detailed descriptions in the retrieved textual descriptions cannot match image object features correctly. For example, with regard to limitation 3, Figure 6c shows that the retrieved result from VSRN and UNITER both mistakenly described too many details about the woman. Limitation 4 is when one part of the textual description relates to the image, but the other part does not match the image. For example, Figure 6d is a result of VSRN with no text in the description for this image about the wedding. Figure 6e shows that UNITER missed ‘people are controlled by officers’ in its textual description. Limitation 5 reveals the VSE network has no accurate concept of the number of main objects in an image. There are two people in Figure 6f, but the results of VSRN and UNITER described three people and one person, respectively. These limitations suggest that learning image details is currently a challenge for VSE networks.

Group 3 generalises that limitations 6, 7, and 8 are related to a network’s ability to extract higher-level visual semantics. For Figure 6g, VSRN and UNITER retrieved the description related to ‘rhythmic gymnasts’ and ‘clutching yellow ski handles by a man’, respectively, an error related to limitation 6. Limitation 7 was derived after observing many irrelevant cases where the retrieved textual descriptions are too simple to describe the rich contents of the image. Taking Figure 6h as an example, the retrieved result by VSRN and UNITER did not describe the man’s smile and his exact action. Limitation 8 shows that VSRN and UNITER did not perform well in retrieving the descriptions of the images which contained human postures and actions. For example, in Figure 6i, the action of ‘roping animal’ is recognised as ‘laying’ by VSRN and UNITER. These limitations suggest that VSE networks are limited in their ability to extract higher and more complex visual semantic information at present.

Group 4 shows the basic functions of neural networks, with object detection (limitation 9) and recognition (limitation 10) also influencing the performance of networks. The small ‘monk’ objects in Figure 6j, an example of limitation 9, were not detected and neither VSRN nor UNITER could give correct descriptions. ‘The man being shaved’ in Figure 6k, an example of limitation 10, was mistakenly recognised as a ‘child’ and ‘woman’ by VSRN and UNITER, respectively.

### 4.3. Discussion on the Strengths and Limitations of VSE Networks

This subsection summarises the strengths and limitations of the VSE++, SCAN, VSRN, and UNITER networks. Figure 7 shows that for 60% of the 2014 image queries, VSRN and UNITER both retrieved the same relevant descriptions at first rank for those queries; and also shows that for 10% of the image queries, VSRN and UNITER retrieved irrelevant descriptions at first rank. However, for 30% of the queries there was no agreement between VSRN and UNITER in the retrieved descriptions.

In the discussion that follows focuses on comparing the attention mechanisms of VSE networks as a strategy for understanding how the attention mechanisms impact their image-to-text retrieval performance. Five attention mechanisms are utilised by the networks for giving attention to important information across images and text. These mechanisms are: (1) Image–text attention aligns image regions and words with crossing modalities; (2) Image–self attention weights the relations between image regions; (3) Text–self attention weights the relations between words; (4) Detailed visual attention weights the detail features in the image object; (5) Global visual reasoning attends to the relations between a group rather than a pair of image objects for reasoning the visual semantics globally.

Table 6 is used for indicating how the attention mechanisms impact the performance (i.e., average Precision@1) for VSE++, SCAN, VSRN, and UNITER. The joint analysis of Table 5 and Table 6 reveals the strengths and limitations of VSE++, SCAN, VSRN, and UNITER. They are described below:VSRN applies a GRU for global visual reasoning based on the pairwise relations between image regions extracted by GCN. The limitations in group 1 indicate that the performance of global visual reasoning for VSRN still needs to be improved. Compared to VSRN, UNITER benefits from the multi-layers of the transformer, and it has overcome limitation 1. However, the limitation of missing key image objects by VSRN and UNITER indicates that global visual reasoning is still a challenging problem for VSE networks.Table 6 shows that none of the networks has the attention mechanisms to achieve detailed visual attention. Misclassified cases of VSRN and UNITER from group 2 limitations reveal that the current VSE networks are not using detailed information for cross-modal information retrieval. However, the matched details between image and text should play a positive role in retrieval, while the unmatched parts should make a negative contribution to matching in further research.VSRN performs image–self attention by using GCN to compute the relations between image regions, so the average Precision@1 of VSRN is 1.8% higher than that of SCAN, as shown in Table 6. UNITER applies transformers to achieve image–self, text–self, and image–text attentions, and it outperformed other networks by more than 11% in average Precision@1. Therefore, this progress shows that the extraction of high-level visual semantics can improve the VSE networks. According to the limitations of group 3, as described in Table 5, there is still a need to improve the extraction of visual semantics for VSRN and UNITER, so higher-level visual semantics are necessary to VSE networks. In addition, SCAN outperformed VSE++ by using the stacked cross attention on image–text, where the average Precision@1 was improved by almost 27.5%. UNITER also uses the transformer for image–text attention, thus cross-modal attention is effective in VSE networks. However, cross-modal attention requires the network to iteratively process image and text pairs, and the retrieval time for 1000 queries of SCAN and UNITER is 187.3 seconds and 4379 seconds, respectively, which is too slow for practice.Group 4 limitations illustrate that VSE networks still need to perfect the basic functions, such as object detection and recognition, of neural networks. At present, the two-stage VSE networks rely on the reliability of the object detection stage.

## 5. Conclusions and Future Work

This study evaluates and compares the performance of four VSE networks, namely VSE++, SCAN, VSRN, and UNITER, for the task of image-to-text retrieval using the Flickr30k dataset. Two experiments were carried out. The first experiment evaluated the retrieval performance of the VSE networks and the second experiment analysed the performance of two of the best-performing VSE networks (i.e., VSRN and UNITER) to determine their limitations. The results of the first experiment revealed that the pre-trained UNITER network achieved the highest retrieval performance across all evaluation measures, followed by the VSRN network. The results of experiment 2 revealed that VSE networks suffer from various limitations which are mainly related to global reasoning, background confusion, attention to detail, and extraction of higher-level visual semantics. Furthermore, the overall retrieval efficiency of the networks needs to be improved for them to be adopted for cross-modal information retrieval tasks, and hence to be embedded in search engines. Understanding the limitations can help researchers advance the area of VSE networks by utilising that knowledge to build future VSE networks that overcome these limitations.

Images can contain various objects and interactions between objects, relative positions of objects, and other high-level semantic concepts, and therefore understanding image content is important in VSE networks for information retrieval [3]. The progress of VSE networks for image–text retrieval is currently determined by comparing the Recall of networks on the Flickr30k public dataset [1,2,3,4]. Most of the work about VSE networks thus far has focused on certain challenges such as image–text alignment [2,4], visual position attention [15], and visual reasoning [3]. To the best of the authors’ knowledge, there is no comprehensive analysis of the limitations of state-of-the-art VSE networks. This study experimentally analyses the performance of VSE algorithms and provides a summary of their limitations from the perspective of image content understanding. Most limitations discussed in this paper can be independently extended to a research direction. The analysis of these limitations will benefit the cross-modal research community and guide future research directions for VSE networks.

Future work includes developing methods for the extraction of higher-level visual semantics based on in-depth relations between image regions, and also developing suitable attention mechanisms that will enable networks to attend to the details of image objects. Future work also includes developing algorithms for improving the efficiency of the pre-trained network which uses the cross-modal attention mechanism and evaluating these networks in real practice. Importantly, there is a lack of research and evaluations of VSE networks when using adversarial data samples. Hence, future work can also include comparing the performance of VSE networks when adopting various perturbation approaches to generate adversarial images and descriptions. Such an analysis can provide an understanding of how adversarial samples can affect the retrieval results of VSE networks, which can aid the development of algorithms and solutions for overcoming the limitations of VSE networks on adversarial samples.

## Figures and Tables

**Figure 1 jimaging-07-00125-f001:**
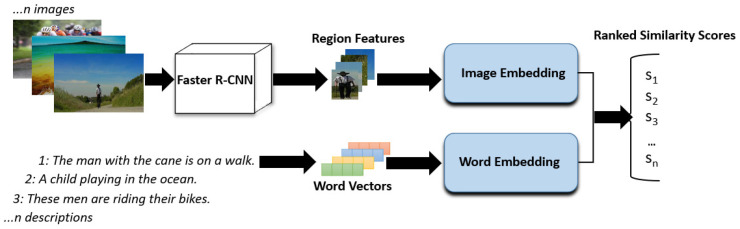
Flowchart illustrating the two-stage process of VSE networks (VSE++, SCAN, PFAN++, VSRN, and UNITER) when applied to the task of information retrieval.

**Figure 2 jimaging-07-00125-f002:**
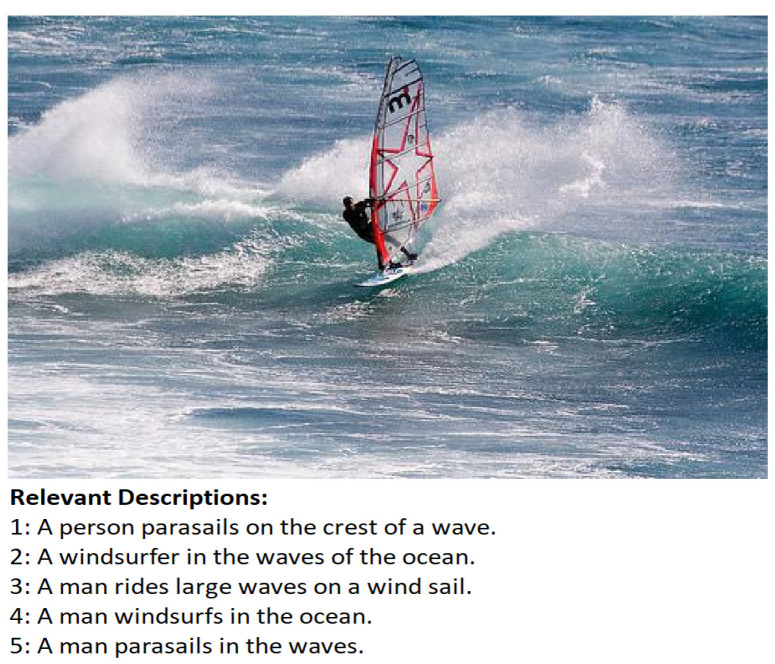
Image example of Flickr30K. One image has five relevant textual descriptions.

**Figure 3 jimaging-07-00125-f003:**
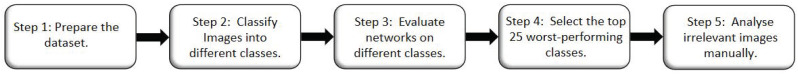
Flow chart illustrating the methodology for experiment 2.

**Figure 4 jimaging-07-00125-f004:**
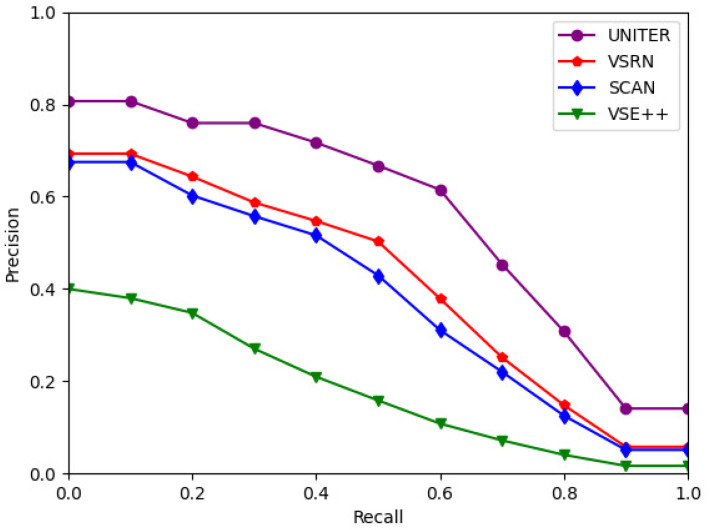
Average PR curves of VSE networks for image-to-text retrieval.

**Figure 5 jimaging-07-00125-f005:**
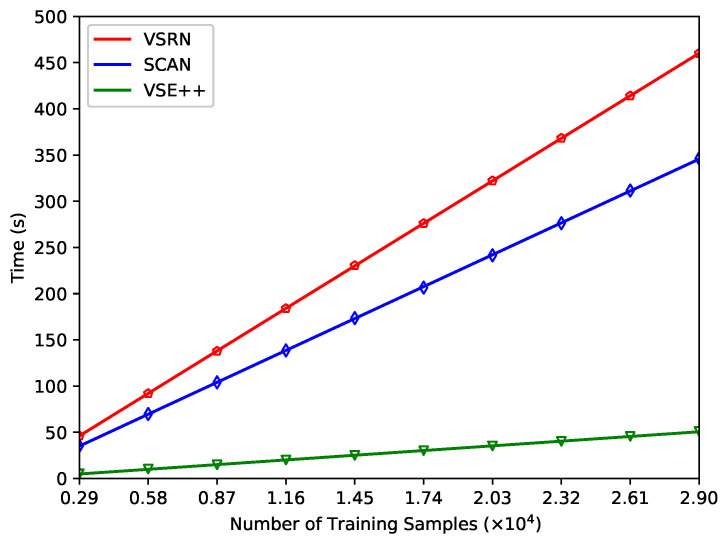
Computation time of VSE networks when using various training samples. Three lines fit the data points which are shown by the triangle, diamond, and pentagon for VSE++, SCAN, and VSRN, respectively.

**Figure 6 jimaging-07-00125-f006:**
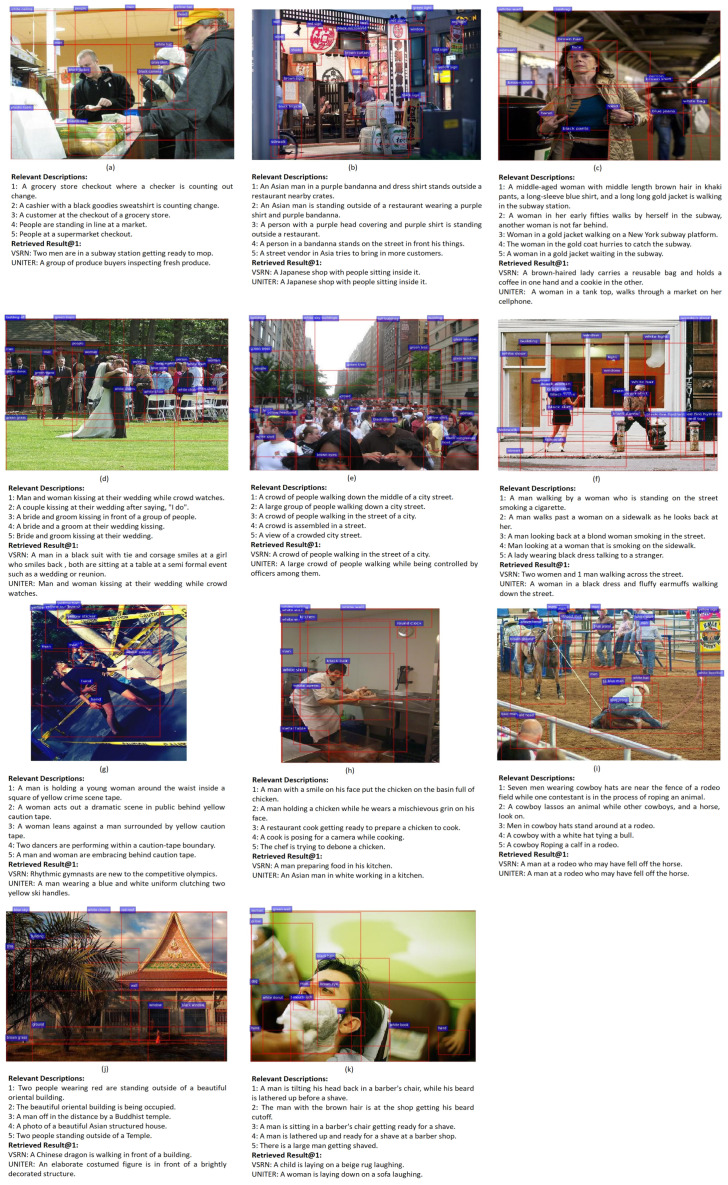
A set of images relevant to the limitations provided in Table 5. One image contains 5 relevant descriptions, and the first retrieved description by VSRN and UNITER is denoted as retrieved result at rank 1 (i.e., Result@1).

**Figure 7 jimaging-07-00125-f007:**
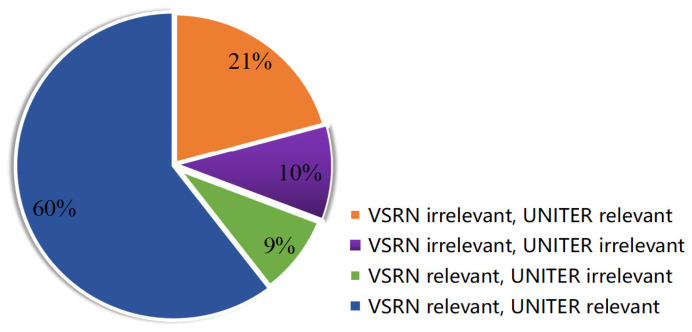
Percentage of relevant and irrelevant descriptions retrieved at rank 1 for 2014 image queries from the Flickr30K dataset.

**Table 1 jimaging-07-00125-t001:** Forty classes with the largest number of images in the query set.

Forty Classes with the Largest Number of Images					
Descriptions	No. of Images	Descriptions	No. of Images	Descriptions	No. of Images
unicycle, monocycle	53	ballplayer, baseball player	21	swimming trunks, bathing trunks	15
stage	50	crutch	19	racket, racquet	14
mountain bike, all-terrain bike off-roader	43	miniskirt, mini	19	sarong	14
restaurant, eating house, eating place, eatery	40	rugby ball	19	volleyball	14
soccer ball	29	library	18	lakeside, lakeshore	14
swing,	27	stretcher	18	sandbar, sand bar	14
ski	26	maypole	17	barbershop	13
jinrikisha, ricksha, rickshaw	23	umbrella	17	prison, prison house	13
cliff, drop, drop-off	22	grocery store, grocery food market, market	16	accordion, piano accordion squeeze box	12
horizontal bar, high bar	21	jean, blue jean, denim	16	croquet ball	12

**Table 2 jimaging-07-00125-t002:** Average Recall values for retrieving any 1 of 5 relevant descriptions. The results were computed by averaging the performance of each model across 1000 queries.

Performance Evaluation of Networks: Retrieving any 1 of 5 Relevant Textual Descriptions				
Network	Average Recall@1 (%)	Average Recall@5 (%)	Average Recall@10 (%)	Average Recall@20 (%)
VSE++	40.0	70.4	80.6	87.3
SCAN	67.5	90.2	94.1	97.1
VSRN	69.3	90.2	94.2	97.1
UNITER	80.8	95.7	98.0	99.0

**Table 3 jimaging-07-00125-t003:** Comparison of average Recall, average Precision, and average $F_{1}$-score between VSE++, SCAN, VSRN, and UNITER for retrieving 5 of 5 relevant textual descriptions. The results were computed by averaging values from all 1000 queries.

Performance Evaluation of Networks: Retrieving 5 of 5 Relevant Textual Descriptions				
@K	Network	Average Recall (%)	Average Precision (%)	Average $F_{1}$-Score (%)
5	VSE++	29.4	29.4	29.4
	SCAN	47.1	47.1	47.1
	VSRN	50.3	50.3	50.3
	UNITER	61.5	61.5	61.5
10	VSE++	41.9	21.0	27.9
	SCAN	62.1	31.1	41.4
	VSRN	64.5	32.3	43.0
	UNITER	76.4	38.2	50.9
20	VSE++	54.7	13.7	21.9
	SCAN	73.9	18.5	29.5
	VSRN	76.0	19.0	30.4
	UNITER	85.5	21.4	34.2
50	VSE++	70.6	7.1	12.8
	SCAN	85.1	8.5	15.5
	VSRN	86.5	8.7	15.7
	UNITER	93.2	9.3	17.0
100	VSE++	80.1	4.0	7.6
	SCAN	90.8	4.5	8.6
	VSRN	92.0	4.6	8.8
	UNITER	96.0	4.8	9.2

**Table 4 jimaging-07-00125-t004:** The 25 worst-performing classes of ImageNet classes by VSRN and UNITER for image-to-text retrieval. Class names are aligned to ease interpretation of results.

Performance on ImageNet Classes Using the Experiment 2 Query Set					
VSRN			UNITER		
Topic (Keyword)	Average Precision@1 (%)	Average Recall@5 (%)	Topic (keyword)	Average Precision@1 (%)	Average Recall@5 (%)
butcher shop, meat market	20.0	14.0	butcher shop, meat market	60.0	32.0
torch	27.3	21.8	torch	45.5	32.7
prison, prison house	53.9	35.4	prison, prison house	61.5	47.7
cowboy hat, ten-gallon hat	54.6	36.4	cowboy hat, ten-gallon hat	72.7	58.2
accordion, piano accordion, squeeze box	33.3	36.7	accordion, piano accordion, squeeze box	75.0	51.7
stretcher	88.9	38.9	stretcher	55.6	43.3
stage	58.0	40.0	stage	76.0	53.2
crutch	63.2	42.1	crutch	84.2	46.3
jean, blue jean, denim	50.0	42.5	jean, blue jean, denim	81.3	57.5
grocery store, grocery, food market, market	43.8	42.5	grocery store, grocery, food market, market	68.8	52.5
volleyball	50.0	44.3	volleyball	85.7	60.0
restaurant, eating house, eating place, eatery	65.0	44.5	restaurant, eating house, eating place, eatery	80.0	57.0
barbershop	53.9	44.6	barbershop	92.3	53.9
maypole	64.7	45.9	maypole	82.4	54.1
jinrikisha, ricksha, rickshaw	69.6	47.8	jinrikisha, ricksha, rickshaw	82.6	60.9
rugby ball	89.5	48.4	rugby ball	79.0	56.8
military uniform	41.7	35.0	football helmet	80.0	46.0
miniskirt, mini	52.6	43.2	basketball	72.7	49.1
pole	83.3	46.7	soccer ball	89.7	55.2
umbrella	82.4	47.1	horizontal bar, high bar	71.4	57.1
unicycle, monocycle	71.7	47.2	moped	80.0	58.0
paddle, boat paddle	70.0	48.0	cinema, movie theater, movie theatre, movie house, picture palace	70.0	58.0
lakeside, lakeshore	71.4	48.6	swing	81.5	59.3
ballplayer, baseball player	81.0	49.5	library	77.8	60.0
fountain	70.0	50.0	swimming trunks, bathing trunks	86.7	61.3

The first part of the table shows the class names found in both the results of VSRN and UNITER, and these are arranged from smallest to largest according to the average Precision@1 values of VSRN. The bottom part of the table shows the rest of the classes arranged in order of smallest to largest according to the average Precision@1 values of VSRN and UNITER, respectively.

**Table 5 jimaging-07-00125-t005:** Summary of 10 limitations of VSE networks. This table is based on the analysis of irrelevant cases from the 25 worst-performing classes of VSRN and UNITER.

Summary of Limitations of VSE Networks		
No.	Limitation Title	Limitation Description
**Group 1: The VSE networks do not globally reason with the image scene (limitation 1 only applies to VSRN, limitation 2 applies to VSRN and UNITER)**		
1	Background issue	The VSE networks cannot accurately recognise the content of the image’s foreground based on its background.
2	Missing key objects	Key objects, which are important to the image content, are ignored.
**Group 2: The VSE networks do not give enough attention to the detailed visual information (all limitations apply to VSRN and UNITER)**		
3	Errors in retrieved descriptions	Details of objects from the retrieved textual descriptions do not match the details of the image.
4	Partially redundant descriptions	Only part of the retrieved textual description is relevant to the image.
5	Object counting error	The networks cannot correctly count objects in an image.
**Group 3: The VSE networks’ capability in extracting the higher-level visual semantics needs to be improved (all limitations apply to VSRN and UNITER)**		
6	Visual reasoning error	The capability for extracting the higher-level semantics for visual reasoning of the VSE networks is inadequate.
7	Imprecise descriptions	Retrieved descriptions do not provide enough detail to describe the rich content of images.
8	Action recognition issue	Actions and postures of objects in retrieved textual descriptions sometimes do not match the image content.
**Group 4: The basic functions, i.e., object detection and recognition, of neural networks need to be improved (all limitations apply to VSRN and UNITER)**		
9	Detection error	Some key objects are missed at the object detection stage.
10	Recognition error	Image object attributes are recognised incorrectly.

**Table 6 jimaging-07-00125-t006:** Comparison of attention mechanisms among VSE networks. The values of average Precision@1 is from experiment 1, average Precision@1 is equal to average Recall@1 of retrieving any 1 of 5 relevant textual descriptions. Retrieval time is the time taken to retrieve 1000 queries in Flickr30K’s test set by the networks when working on the hardware of an NVIDIA GEFORCE RTX 2070 graphics card.

Comparison of Attention Mechanisms Amongst VSE Networks							
Network	Image–Text Attention	Image–Self Attention	Text–Self Attention	Global Visual Reasoning	Detailed Visual Attention	Average Precision@1 (Image-to-Text)	Retrieval Time
VSE++	None	None	None	None	None	40.0%	4.4s
SCAN	Stacked Cross	None	None	None	None	67.5%	187.3s
VSRN	None	GCN	None	GRU	None	69.3%	13.6s
UNITER	Transformer	Transformer	Transformer	Transformer	None	80.8%	4379s

## Data Availability

The code for the experiments presented in this paper can be found in the project’s GitHub repository https://github.com/yangong23/VSEnetworksIR (accessed on 25 July 2021).

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
