# Peer review of "On the Limitations of Visual-Semantic Embedding Networks for Image-to-Text Information Retrieval"

_2313-433X, 2021, doi:10.3390/jimaging7080125_

Round 1
Reviewer 1 Report
This paper analyses the current visual-semantic embedding (VSE) networks for information retrieval in-depth with various methods and summarizes the limitations of them. Future research was pointed by the paper through the comparison of strengths and limitations of the current VSE networks. These analyses explore the essence of how VSE networks work, that is important to promote the VSE network to be used in the practice. This paper seems to be good with solid contribution, and my comments are as follows:
- Please explain why it is necessary to evaluate the performance of models on retrieving all 5 descriptions since that on retrieving any 1 of 5 descriptions has been evaluated.
- Why the Recall/Precision curve in Figure 4 is needed since that F1-measure has been computed through Recall and Precision.
- Please explain how the classes of UNITER in Table 4 align with VSRN.
- Can ‘Objects Quantity Error ’ of title 5 in Table 5 be replaced by ‘Objects Counting Error ’?
Author Response
We are grateful to the Reviewers for their positive and constructive comments that helped us improve the quality of the manuscript. Please see the attachment.

Reviewer 2 Report
The paper just compares visual-semantic embedding networks without any additional contributions. The paper finds limitations but does not provide empirical studies on some proposed solution. I would strongly recommend the authors to address these limitations and come up with a proposed method
The authors should also perform experiments on sample complexity: How these methods perform for different amount of training samples ?
It would be great for the authors to address retrieval results on adversarial images. How are the models robust in retrieval results of adversarial images ?
Author Response

(The authors gave the same response as above.)

Round 2
Reviewer 2 Report
The paper has been improved but not state-of-the-art technique. Still it is fine enough for this journal.